# Accuracy of Machine Learning Classification Models for the Prediction of Type 2 Diabetes Mellitus: A Systematic Survey and Meta-Analysis Approach

**DOI:** 10.3390/ijerph192114280

**Published:** 2022-11-01

**Authors:** Micheal O. Olusanya, Ropo Ebenezer Ogunsakin, Meenu Ghai, Matthew Adekunle Adeleke

**Affiliations:** 1Department of Computer Science and Information Technology, Sol Plaatje University, Kimberley 8300, South Africa; 2Biostatistics Unit, Discipline of Public Health Medicine, School of Nursing & Public Health, College of Health Sciences, University of KwaZulu-Natal, Durban 4000, South Africa; 3Discipline of Genetics, School of Life Sciences, University of KwaZulu-Natal, Durban 4000, South Africa

**Keywords:** diagnosis, soft computing, predictive models, type 2 diabetes mellitus, meta-analysis

## Abstract

**Highlights:**

We reviewed soft-computing and statistical learning methods for predicting type 2 diabetes mellitus.We searched for papers published between 2010 and 2021 on three academic search engines, obtaining 34 relevant documents for the final meta-analysis.We analyzed the data extracted, compared the results and models, discussed their performance, and highlighted the issues related to T2DM.Finally, the decision trees model has the best prediction performances, with excellent accuracy compared to other soft-computing models in this systematic meta-analysis.

**Abstract:**

Soft-computing and statistical learning models have gained substantial momentum in predicting type 2 diabetes mellitus (T2DM) disease. This paper reviews recent soft-computing and statistical learning models in T2DM using a meta-analysis approach. We searched for papers using soft-computing and statistical learning models focused on T2DM published between 2010 and 2021 on three different search engines. Of 1215 studies identified, 34 with 136952 patients met our inclusion criteria. The pooled algorithm’s performance was able to predict T2DM with an overall accuracy of 0.86 (95% confidence interval [CI] of [0.82, 0.89]). The classification of diabetes prediction was significantly greater in models with a screening and diagnosis (pooled proportion [95% CI] = 0.91 [0.74, 0.97]) when compared to models with nephropathy (pooled proportion = 0.48 [0.76, 0.89] to 0.88 [0.83, 0.91]). For the prediction of T2DM, the decision trees (DT) models had a pooled accuracy of 0.88 [95% CI: 0.82, 0.92], and the neural network (NN) models had a pooled accuracy of 0.85 [95% CI: 0.79, 0.89]. Meta-regression did not provide any statistically significant findings for the heterogeneous accuracy in studies with different diabetes predictions, sample sizes, and impact factors. Additionally, ML models showed high accuracy for the prediction of T2DM. The predictive accuracy of ML algorithms in T2DM is promising, mainly through DT and NN models. However, there is heterogeneity among ML models. We compared the results and models and concluded that this evidence might help clinicians interpret data and implement optimum models for their dataset for T2DM prediction.

## 1. Introduction

Data mining, such as soft-computing (that is, machine learning (ML)) methods, has become essential in diagnosing T2DM and assigning management to healthcare providers [1]. ML is a subdivision of artificial intelligence that is gradually exploited within the field of diabetic medicine. Primarily, it is how computers make sense of data and classify tasks with or without human supervision. Several ML models have been used extensively in diabetes mellitus (DM) studies to explore DM risk factors [2,3]. The ML methods, which include logistic regression (LR), artificial neural networks (ANN), and decision trees (DT), were used to predict both DM and pre-diabetes [4,5,6,7,8,9]. Other ML models, such as random forest (RF), support vector machines (SVM), k-nearest neighbors (KNN), and the naïve Bayes, have also been used in the literature [10,11,12,13,14,15].

Given the above, previous studies have distinct pragmatic shreds of evidence for each ML model [16,17]. Still, no agreement has arisen to guide the choice of precise ML models for clinical investigation in the context of diabetic medicine. The overall classification accuracy reported in each model varies from one study to another. Furthermore, ML studies conveyed the model evaluation criteria, including the area under the curve (AUC). Most significantly, an adequate boundary for AUC to be employed in clinical investigation and suitable ML models that are efficient in diabetic research have yet to be appraised. As a result of the visible success in a wide range of predictive tasks, medical researchers and clinicians have a significant interest in using ML techniques.

Nevertheless, pooled estimates for ML techniques among patients with T2DM and the trends over the years remain unknown globally. Against this background, our study pooled data from previous independent studies to determine the overall ML models’ predictive ability of T2DM disease. Our findings could be helpful to clinicians, healthcare managers, and policymakers involved in the delivery of Type-2 diabetes healthcare worldwide.

## 2. Materials and Methods

### 2.1. Search Strategy and Selection Process

In this meta-analysis, we searched the scholastic databases of Web of Science, Scopus, and PubMed for relevant published articles on ML applied to health applications for T2DM. These databases were searched for English papers published between 2010 and 2021. We excluded studies published before January 2010 because most of those studies used outdated computer-aided algorithms that are currently not popular. The literature search strategy, selection of publications, data extraction, and reporting results were executed following the Preferred Reporting Items for Systematic reviews and Meta-Analyses (PRISMA) (Moher et al., 2010). During a comprehensive literature search, the search terms used were: With MESH terms: (“diabetes mellitus, type 2” [All Fields] AND (“machine learning” [All Fields] OR “deep learning” [All Fields] OR “neural networks (computer)” [All Fields] OR “support vector machine” [All Fields] OR “classification” [All Fields] OR “decision trees” [All Fields] OR “cluster analysis” [All Fields] OR “principal component analysis” [All Fields] OR “data mining” [All Fields] OR “logistic models” [All Fields] OR “algorithms” [All Fields])) AND (“diagnosis” [All Fields] OR “roc curve” [All Fields] OR “area under curve” [All Fields])”, (“machine learning” or “deep learning” or “artificial intelligence” or “neural network” or “support vector machine” or “classification-tree” or “regression-tree” or “decision-tree” or “random forest” or “gradient boosting” or “k-nearest neighbors” or “supervised-learning” or “unsupervised-learning”. The search terms were separated or combined using Boolean operators such as “OR” or “AND”. After data extraction, we summarized and reported the findings in tables and figures according to the study’s objectives.

### 2.2. Inclusion Criteria

The inclusion criteria were original articles and clinical trials. In addition, those studies with model performance evaluation, such as accuracy, sensitivity, specificity, and area under the curve (AUC), were included.

### 2.3. Exclusion Criteria

Articles written in languages other than English, published before January 2010, or with study designs such as reviews, letters to editors, editorials, commentaries, expert opinions, books, book chapters, brief reports, and theses were excluded. Conference articles, grey literature, and literature that failed to report model performance evaluation criteria were excluded.

### 2.4. Assessments of Methodological Quality

The quality of the individual studies was independently assessed based on the Quality Assessment of Diagnostic Accuracy Studies (QUADAS-2) tool. QUADAS-2 is a validated tool used to evaluate the quality of diagnostic accuracy studies by patient selection, index tests, reference standards, and the risk of bias for internal and external validity for applicability concerns of individual studies. In this meta-analysis, each article’s qualities were evaluated. The two authors assessed the identified methodological quality and eligibility of articles, and disagreements among reviewers were fixed accordingly with discussion. The data extracted included the author, the publication year, the country where the study was conducted, the study design, the sample size, the prediction type, the T2DM cases, the number of participants, the sensitivity, the specificity, the impact factor of the articles (extracted from Scopus webpage), the ML models and the software deployed. In addition, we included the model that had the best overall performance in the primary analysis for the studies that proposed multiple models. We also extracted the performance of models with the best sensitivity and specificity in studies with numerous ML models to perform further sensitivity-focused and specificity-focused analyses.

### 2.5. Statistical Analysis

A meta-analysis was conducted for the pooled overall classification accuracy proportion; a chi-squared test was used for heterogeneity; Higgins I-squared (I^2^) was used to assess the total heterogeneity/total variability among studies. For Higgins I-squared (I^2^) [18,19], forest plots of over 50% were observed as an indication of heterogeneity among studies. If the estimated amount of total heterogeneity (Tau I^2^) (DerSimonian and Laird, 1986) was less than 40%, the studies were considered similar. Because the extracted articles were from general populations, a random-effects meta-analysis was deemed to be taken from an inverse-variance model [20].

Additionally, a subgroup analysis was performed to investigate the heterogeneity among the studies based on the prediction type of the algorithm for T2DM and machine-learning diabetes prediction. Combining only the published studies may lead to an insignificant or biased result in the meta-analysis. Thus, this study used a funnel plot to report publication bias among the included studies [18]. The publication bias was assessed through the Begg & Eggers test and the visual inspection of the funnel plot. Meta-regression was used to explore the factors possibly contributing to the between-study heterogeneity. The extracted data were captured into an excel spreadsheet. A meta-analysis was performed via the metafor, rma, meta, and metaprop packages in R (version 4.0.3, R Core Team, Vienna, Austria); the statistical significance was expressed with a 95% Confidence Interval (CI), and *p*-values < 0.05 were considered statistically significant.

## 3. Results

### 3.1. Characteristics of Selected Studies

The features of the eligible studies in Table 1 showed that the application for T2DM with most of the included studies was diagnostic (38.2%, 13/34), followed by prognostic (26.5%, 9/34), nephropathy (20.6%, 7/34), screening and diagnosis (8.8%, 3/34) and risk factor analysis (5.9%, 2/34). The learning algorithm subset of artificial intelligence includes all the methods and algorithms that enable the machines to automatically learn mathematical models to extract useful information from large datasets. Thus, in terms of learning algorithm classification techniques, 23.53% (8/34) of studies applied linear regression (LR), and 23.53% (8/34) used decision trees (DT) on the diabetes patient’s data, respectively. A total of 17.65% (6/34) applied an artificial neural network (ANN), 8.82% (3/34) deployed random forest (RF), and 14.71% (5/34) employed a support vector machine (SVM). One (2.94%) example of a hybrid model, a neural network (NN), a CRISP method, and phenotyping, respectively.

### 3.2. Meta-Analyses Methods

The literature search of three databases (Web of Science, PubMed, and Scopus) and reference screening yielded 1215 articles. We imported all the retrieved articles into EndNote X9, identifying 945 duplicates. Out of the remaining documents, 98 were excluded because their abstracts and titles did not meet the eligibility requirements. Additionally, 172 studies were eligible for a full review, out of which 130 were excluded for not reporting the outcome variable, incomplete information, or non-relevant. A total of 42 studies were eligible for quality assessment, and, finally, 34 documents were found to qualify and were included in the final meta-analysis (Figure 1). The flow diagram in Figure 1 summarizes the reasons for excluding research articles from study inclusion following the PRISMA.

### 3.3. Spatial Distribution of Articles and Soft-Computing Models

Machine learning techniques are popular compare to other methods due to their outstanding classification performance. The distribution of articles by year of publication is shown in Figure 2a. It was evident that publications related to the application of ML techniques in diagnosing diabetes mellitus increased significantly from 2013 to 2016. Based on the inclusion criteria, we also noted a downward trend for publications in the past four years. Many factors could be attributed to this downward trend, but we can only attribute this observation to the inclusion criteria and the disease under investigation in the current study. Thus, we cannot generalize since other researchers can apply the techniques to other diseases.

Additionally, Figure 2b shows the frequency of algorithms applied specifically in ML. Based on the articles that met the inclusion criteria, decision trees are the most significantly used ML techniques in predicting T2DM. It can be said that the four most popular ML models are LR, ANN, DT, and SVM, consecutively.

Given the data sources for the included articles, Figure 3a shows the trend between the impact factor and the publication year. There were 22 studies released between 2013 and 2016 (65%). Algeria and Japan each contributed to one study (medium impact factor = 3.06 and 2.78, respectively); China and the United States each contributed to nine and five studies (medium impact factor = 2.71 and 3.03, respectively). In addition, a substantial impact study was conducted in the Netherlands [34], and another was conducted in Denmark [24]. A moderate impact factor was undertaken in Germany [5], and Brazil and Iran each contributed to a lower impact factor. Figure 3b gives an exhaustive comparison of regional differences in publications and the country’s average impact factor.

Furthermore, Figure 4 shows the frequency of ML applications to health aspects of T2DM. The results showed that the most common medical application of ML for T2DM care was diagnostics, with a 38% frequency, followed by prognostics (26%). 

### 3.4. Results of the Meta-Analysis

#### Proportions of Classification Accuracy

As acknowledged earlier, the meta-analysis results were based on the 34 documents that met the inclusion criteria. The summary proportion was presented as a random effect due to the heterogeneity of estimates across studies. This classification accuracy was 86% (95% CI: 82–89%). The I^2^ was 99.00% (95% CI: 99.54–99.84%) of the total variance between studies. A possible reason for this high heterogeneity could be attributed to the sampling error between studies and other design aspects. Tau I^2^ was 59% (95% CI: 0.39–1.13%) (SE = 0.1128). The Q statistic Q (df = 33) = 4202.3722, *p*-value < 0.0001, which indicated that the included studies did share a standard effect size (Figure 5). So, we concluded that our analysis had substantial homogeneity (Figure 5). 

### 3.5. ML Models and Diabetes Prediction

Machine learning approaches became a standard solution for processing big data analytics when the scope of theoretical knowledge of a problem is incomplete [42] and when the preliminary statistical data are unknown [43]. Because of these factors, combined with their robustness as one of the best techniques to solve non-linear geo-environmental problems, ML techniques are increasingly used in disease forecasting. In addition, different varieties exist within an ML model, and their performance varies depending on the area under investigation and the input data. Due to variations in sample size, studies, inclusion criteria, and methodology, heterogeneity examination in meta-analyses becomes inevitable. Classification diabetes prediction significantly differed between diabetes predictions. It was greatest among models with a screening and diagnosis (*p* = 3, proportion = 0.91, 95% CI [0.74, 0.97]) when compared to nephropathy (*p* = 7, proportion = 0.88, 95% CI [0.83, 0.91]), prognostic (proportion = 0.84, 95% CI [0.77, 0.90]), diagnostic (proportion = 0.84, 95% CI [0.77, 0.89]) and risk factor analysis (proportion = 0.84, 95% CI [0.76, 0.89]) (Figure 6). 

### 3.6. ML Models and Prediction of T2DM

In recent years, ML models have been more widely and increasingly applied in biomedical fields. However, given their complexity and potential clinical implications, there is an ongoing need for further research on their accuracy. The prediction performance of each soft computing approach was compared by using either the accuracy or the area under the curve (AUC) of the receiver operating characteristic curve. Based on the systematic literature, the articles that met the inclusion criteria reported the following algorithms for the prediction of T2DM: DT, hybrid model, LR, NN, phenotyping, RF and SVM, classification algorithm and combined the prediction of them into one to-increase the prediction accuracy of the algorithm. Moreover, for the prediction of T2DM, the DT and ANN models had a pooled accuracy of (*p* = 8, proportion = 0.88, 95% CI [0.82, 0.92]) and (*p* = 6, proportion = 0.85, 95% CI [0.79, 0.89]), resulting in the best approaches in these meta-analyses, respectively. We believe these findings could represent an encouraging step toward the translation to clinical prediction, diagnosis, and prognosis (Figure 7).

Additionally, according to the “no free lunch” theorem (Wolpert et al., 1995), no single learning algorithm universally performs best across all domains. As such, several models should be tested and compared. Thus, these approaches mentioned above were further classified into a linear or non-linear model for straightforward interpretation. The purpose of this section was to compare the classification performance of linear and non-linear ML models for the prediction of T2DM. Overall, non-linear ML models outperformed linear models for the prediction of T2DM (Figure 8). This valuable relative performance information can help researchers select an appropriate non-linear ML model for their studies.

### 3.7. Moderator Analysis

The meta-regression analysis in Table 2 shows that the categorical variables affirmed that the publication year and impact factor did not affect variance in the pooled estimates of classification accuracy. Application for T2DM did not significantly moderate the pooled estimates of classification accuracy, explaining 5.52% of the variance in the pooled classification accuracy proportions (Q_M_ = 2.24, df = 3, *p* = 0.6923; Q_E_ = 3941.8090, df = 29, *p* < 0.0001). Additionally, the model types significantly moderated the pooled estimates of classification accuracy, explaining 46.76% of the variance in the pooled classification accuracy proportions (Q_M_ = 26.04, df = 8, *p* = 0.0010; Q_E_ = 2473.3453, df = 25, *p* < 0.0001). The moderation effects of the model types were driven mainly by the NN and phenotyping subgroup, which accounted for an average total variance in the observed proportions (NN: *β* = 2.36, *p* = 0.0004) and (phenotyping: *β* = −1.40, *p* = 0.0196), respectively. None of the other model types’ subgroups were statistically significant (Table 2). However, the combined model, publication year, impact factor, and application for T2DM and model types explained more heterogeneity (I^2^ = 98.49%, *p* = 0.007, and R^2^ = 54.61%). The pooled classification accuracy proportions decreased insignificantly with an increasing publication year (*p* = 0.5001) and sample size (*p* = 0.1540) (Figure 9 and Figure 10).

### 3.8. Evaluation of Publication Bias

A funnel plot was generated to explore the potential for publication bias. We detected no potential publication bias based on the symmetric shape of the funnel plot of the pooled model performance (Figure 11) and the Eggers’ regression test’s non-significant value (slope = 0.253, *p* = 0.196). Two studies (2, 6) were identified as outliers with a cut-off of (>z^2^), and the Baujat plot showed that there was no single study that influenced the results, and each point represents the number of studies (Figure 12).

## 4. Discussion

### 4.1. Synopsis of Evidence

In recent years, information technologies such as ML models have become essential in predicting T2DM in patients and assigning management to healthcare providers. A significant research focus has been on developing intelligent digital health interventions. To our knowledge, this is the foremost and largest innovative systematic meta-analytic approach in ML model research at a global level, which drew from a wide-ranging number of articles that included over one thousand participants reporting the ML model’s prediction in T2DM disease. In this study, we evaluated the predictive performances of studies using ML prediction models for T2DM. Primary articles were chosen from the Web of Science, Scopus, and PubMed research databases. ML techniques, mixed with other perceptions presented in the learning healthcare systems method, tend to bring better care and management of T2DM to the well-being of society.

Nevertheless, when presenting novel prediction models, one should consider the predictive performance, where the strengths and weaknesses of the ML approaches need to be considered. Recently, numerous modeling methods have been used to predict T2DM and manage T2DM; thus, selecting the most appropriate ML approaches for a specific problem one is trying to solve is always challenging. In this study, we pooled various ML approaches used in previous studies related to T2DM and compared their performance in terms of accuracy. However, the publication year and impact factor did not moderate the aggregate estimates of overall classification accuracy in the meta-regression analyses. However, it is essential to note that our research was limited to the English language. The pooled models’ performance predicted T2DM with an overall accuracy of 86% (95% CI: 82%, 89%), similar to the 82% pooled therapeutic outcomes in depression reported recently [44]. The current pool is slightly higher than the overall c-index of 81.2% reported from a meta-analysis study of use and performance for diabetes prediction in a local setting [45]. This disparity could be attributed to differences in the burden of the disease across study settings, the sensitivity of the diagnostic assays used during these two different periods, and the choice and characteristics of study subjects. High predictive performance was achieved by all models, with accuracy ranging from 0.58 to 0.98. Compared to other models, the DT model performed the best, with an accuracy value of 0.88 (95% CI 0.82–0.92). However, this finding is not surprising because previous studies have revealed that the same ML model can produce diverse accuracy outcomes for the same dataset by selecting various values for the underlying hyperparameters [46,47]. Previous studies have demonstrated the significant role of the DT approach in other medical fields, such as therapeutic outcomes in depression [44] and cardiovascular diseases [48] and predicting diabetes mellitus [49]. Our results confirmed the outstanding performance of the DT method in the risk assessment of T2DM.

Additionally, we grouped the various ML models into three categories: linear, non-linear, and ensemble. The models that used non-linear algorithms to predict T2DM performed better (0.88, 95% CI 0.84–0.91) than the linear model and ensemble modeling approach. This finding is consistent with the previous comparison between linear and non-linear models for classifying thyroid modules [50]. In addition, we also observed that the models based on ML for prediction in T2DM had been mainly focused on screening and diagnostics (0.91, 95% CI [0.74, 0.97]). This observation is also supported by the previous meta-analysis that utilized the ML model for therapeutic outcomes in depression [44]. A possible reason for this finding could be the variation in the year of publication. Our results show a broad spectrum of applications of ML models dominated by predictive approaches.

### 4.2. Policy Implications

Since the discovery of non-infectious diseases, many scientific publications have been produced globally. The current T2DM offers a wide-ranging analysis of the research trends linked to T2DM through documents indexed in academic databases. At the same time, the findings from this systematic survey and meta-analysis have significant policy implications for evaluation and monitoring. These are adequate resources for clinicians to determine if an individual will develop type 2 diabetes mellitus in the coming time. Additionally, synthesizing individuals with T2DM is essential in assisting clinicians in designing an appropriate mechanism to protect vulnerable individuals and reduce pressure on health systems. The current ML techniques have outclassed conventional risk models in predicting T2DM. Still, individuals should be careful about changing their attitude regarding future diabetes risk after having the outcomes of a diabetes prediction test through ML techniques. In addition, ML techniques are vital to improving the predictive capacity of T2DM. Ongoing work should be carried out to build additional precise ML techniques other than the existing ones, supposing that the practicability of utilizing ML in a clinical situation would be improved compared to regular costly and time-consuming blood tests. Finally, the pooling of independent studies gives policymakers the information needed to make informed decisions in uncertain circumstances.

### 4.3. Limitations of the Overview Study

A wide-ranging literature search and watchful data extraction were conducted to avoid bias. However, limitations exist in our study. This systematic meta-analysis was limited to articles written in the English language. In addition, only articles written between 2010 and 2021 were included in the study. Secondly, the authors may have overlooked some valuable keywords and bibliographic sources that may contain relevant articles. Furthermore, due to the scarcity of primary studies, very few preliminary studies have been included to aggregate the accuracy of predictive models at the global level. As a result, in the future, the scope of the study may be broadened to reflect such limitations.

### 4.4. Concluding Remarks and Recommendations

This paper provided an in-depth study of automated T2DM prediction models. It reveals how the data mining and meta-analysis approach can be efficiently implemented in clinical medicine to obtain models that use patient-specific information to predict the end product. Critical articles were compiled from the Web of Science, Scopus, and PubMed scientific repositories. The classification models predicted outcomes for patients diagnosed with T2DM in previously published documents (*p* = 34, n = 136, 952), with an overall accuracy of 86%. The pooled estimates of classification accuracy differed significantly from model to model based on applying the algorithm to T2DM (*p* < 0.01). Predictive models with screening and diagnostics had the most significant overall classification accuracy (pooled proportion = 0.91) compared to models with other algorithms for T2DM (proportion = 0.84 to 0.88).

In summary, our results on the aggregate estimates of model performance can help researchers and decision-makers undertake health technology assessments for various T2DM screening strategies. Hopefully, this analysis will benefit researchers involved in DM therapy’s detection, diagnosis, self-management, and personalization. Additionally, the findings can provide an exhaustive overview of the relative performance of diverse variants of ML models for disease prediction. The implication is that it can aid researchers in selecting appropriate ML models for their studies. Finally, we recommend comparing different ML models to develop a predictive model based on our meta-analysis.

## 5. Conclusions

We pooled data from previous independent studies to determine the overall ML models’ predictive ability of T2DM disease. This systematic review and meta-analysis show that ML models can correctly predict T2DM with good discrimination. Our findings indicated that the decision trees model has the best prediction performances, with excellent accuracy compared to other soft-computing models in this systematic meta-analysis. Moreover, this finding suggests that ML algorithms have a high capacity for advanced enhancement of predictive ability for T2DM. The results are expected to further the global research agenda, and policymakers could use the findings to strengthen medical policies in the clinical diagnosis of a patient with T2DM. This calls for the development of informing procedures for ML for intensive care medicine.

## Figures and Tables

**Figure 1 ijerph-19-14280-f001:**
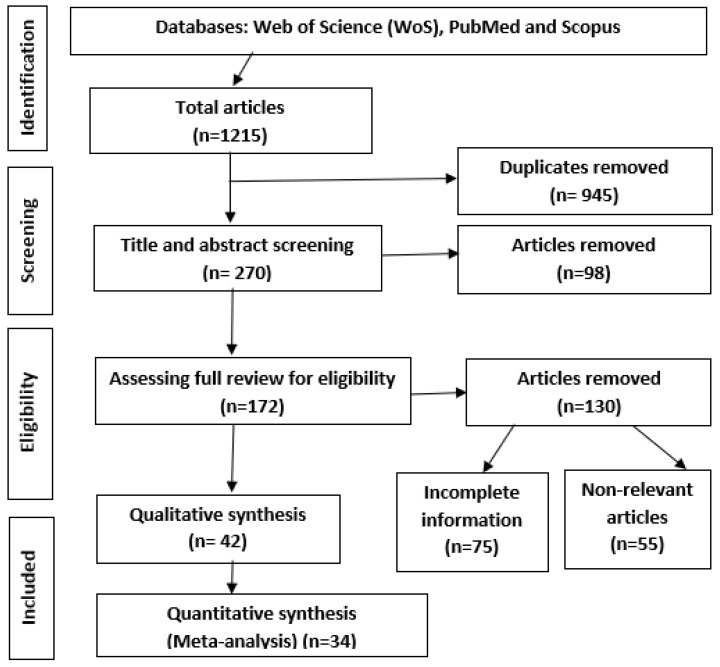
The process of selecting published literature according to *PRISMA* and Meta-Analyses guidelines.

**Figure 2 ijerph-19-14280-f002:**
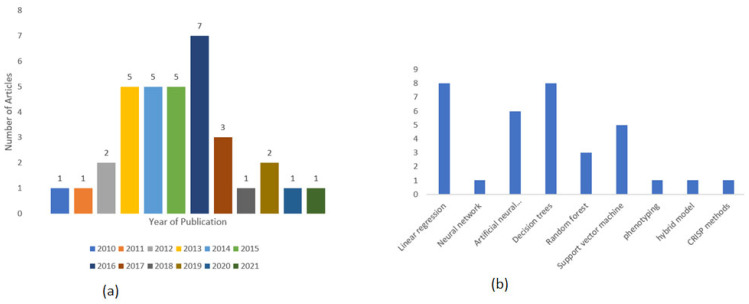
Distribution of articles and soft computing models in meta-analyses: (**a**) Classification of articles by year of publication; (**b**) Soft computing models in general.

**Figure 3 ijerph-19-14280-f003:**
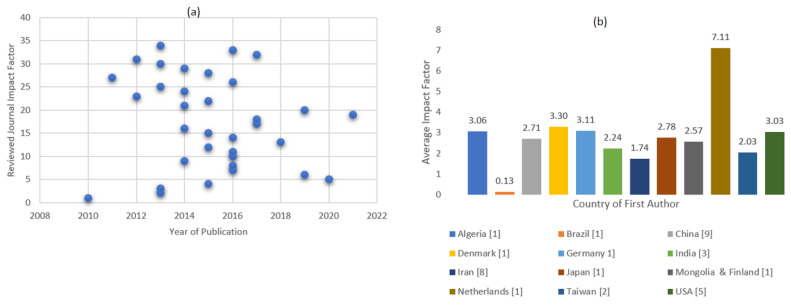
Graphical representation of the temporal and regional trends: (**a**) Temporal trends in publications; (**b**) Regional trends in publications.

**Figure 4 ijerph-19-14280-f004:**
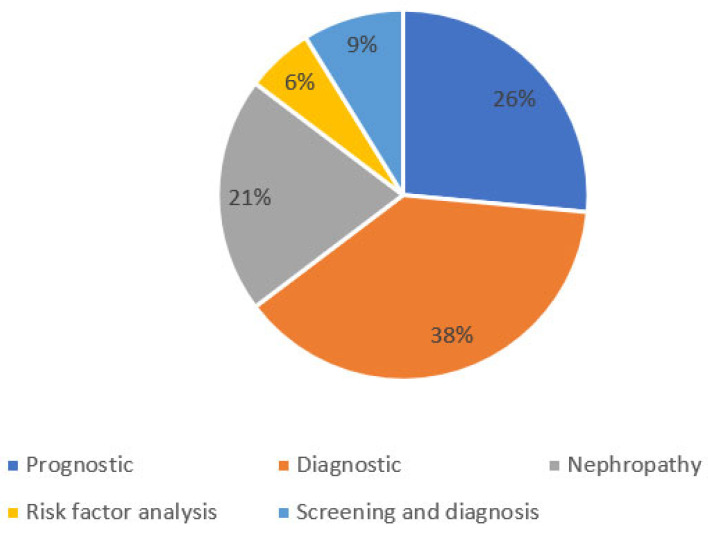
Frequency of machine learning applications for health aspects of type 2 diabetes mellitus.

**Figure 5 ijerph-19-14280-f005:**
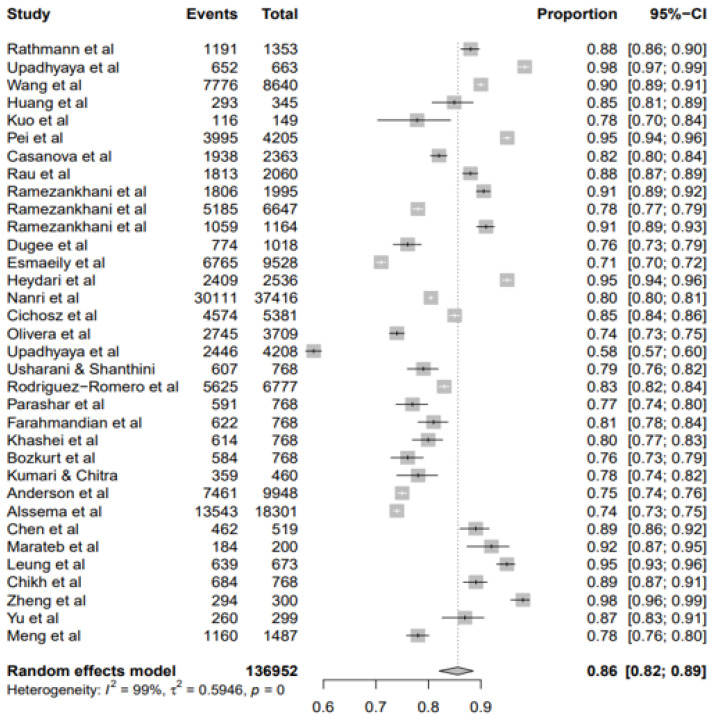
Forest plots showing the proportion of classification accuracy ML models for T2DM.

**Figure 6 ijerph-19-14280-f006:**
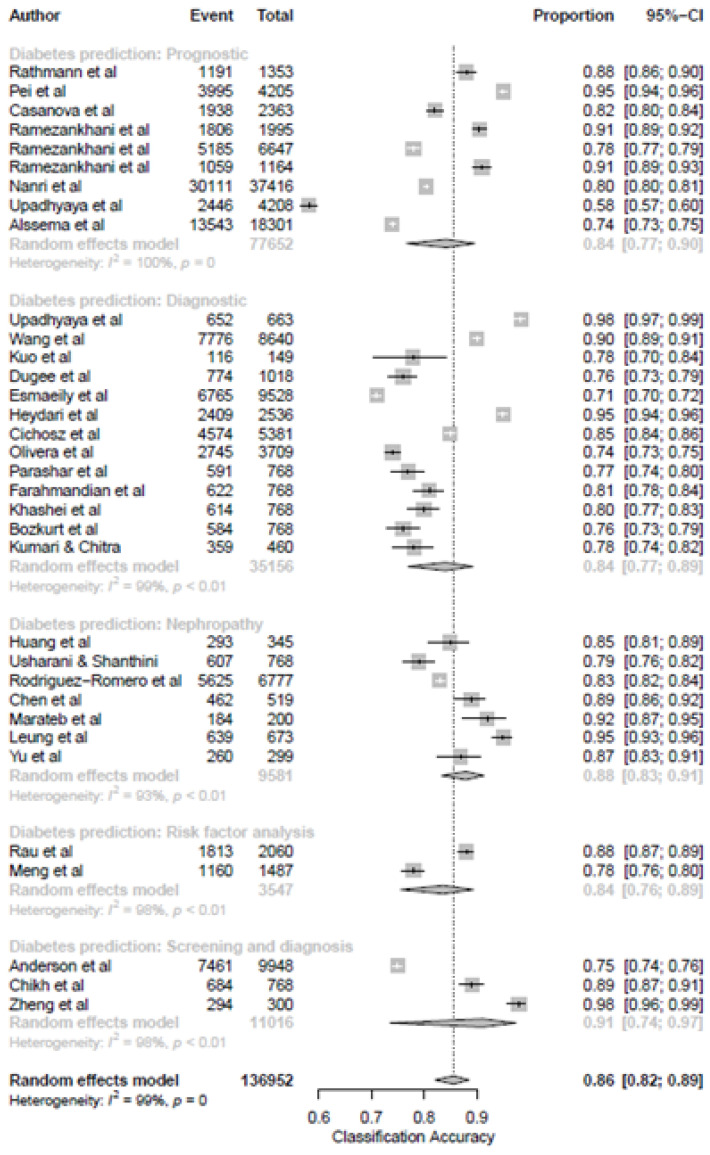
Subgroup analysis of classification accuracy proportions reported by studies that applied a machine learning model to predict type 2 diabetes mellitus.

**Figure 7 ijerph-19-14280-f007:**
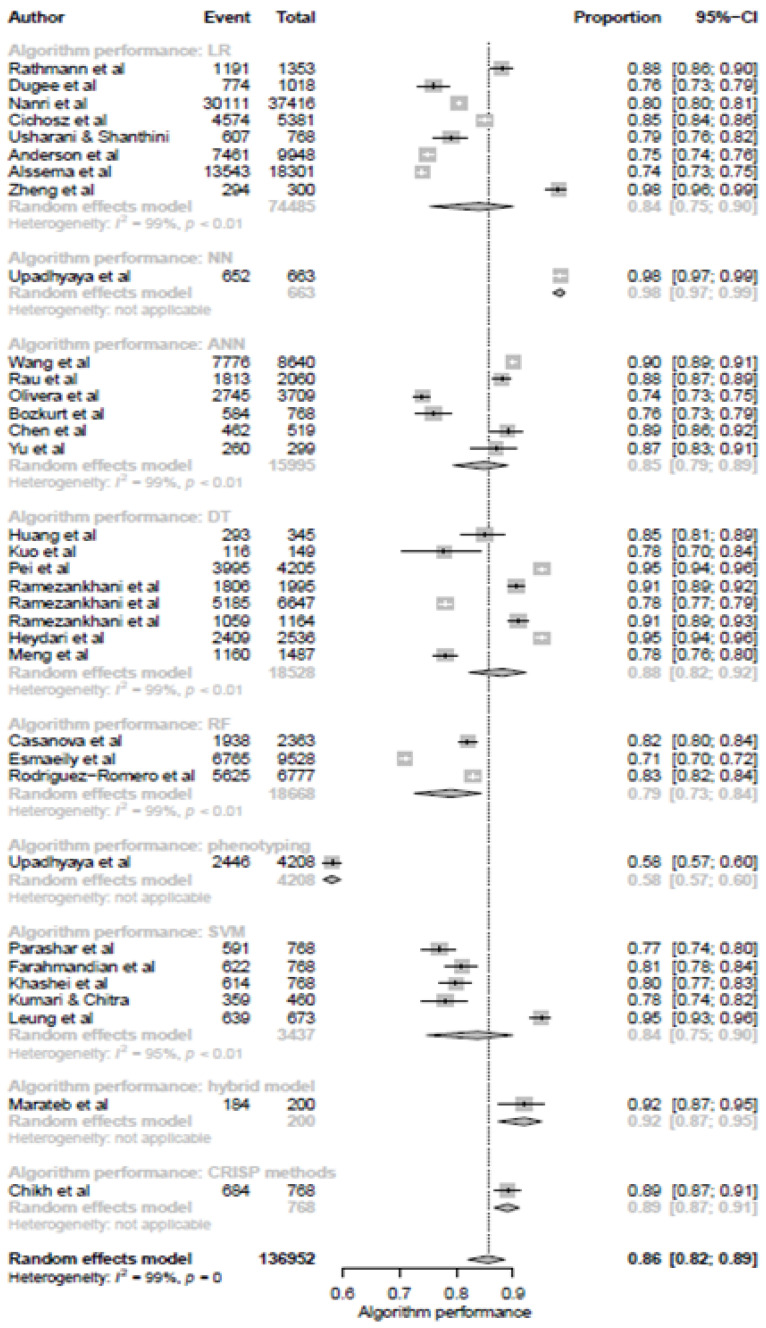
Subgroup analysis based on the various machine learning models for predicting type 2 diabetes mellitus.

**Figure 8 ijerph-19-14280-f008:**
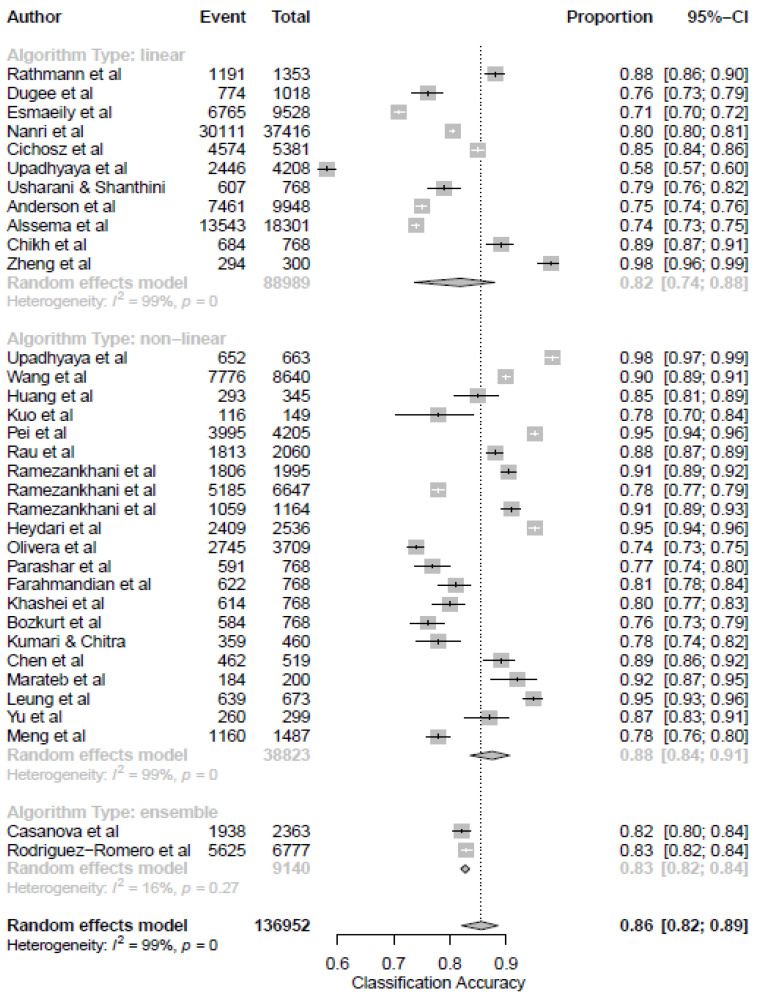
Subgroup analysis based on the machine learning model’s performance in predicting the diagnosis of type 2 diabetes mellitus.

**Figure 9 ijerph-19-14280-f009:**
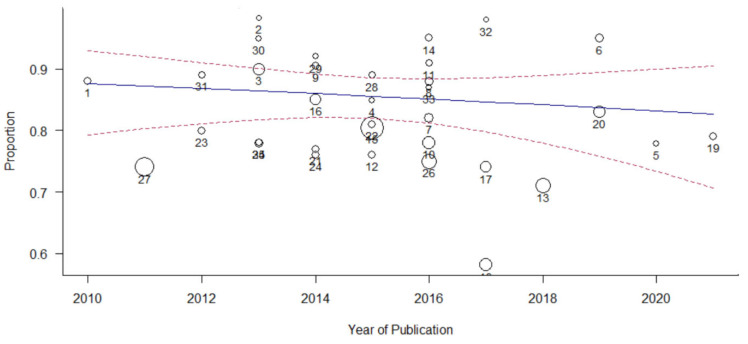
Meta-regression of the performance of the machine learning model in predicting type 2 diabetes mellitus. The study displays the observed effect sizes of the individual studies against the continuous variable publication year.

**Figure 10 ijerph-19-14280-f010:**
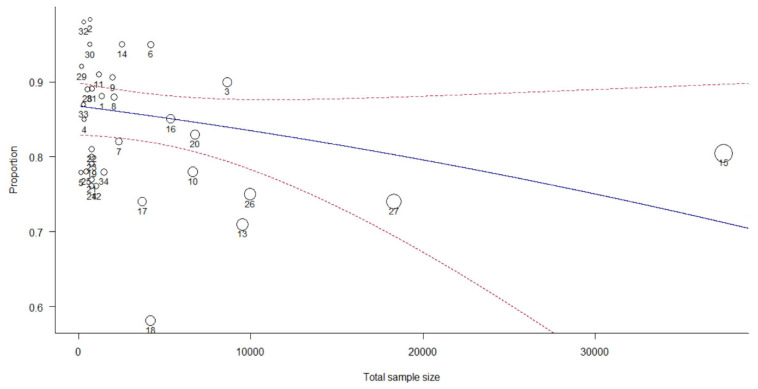
The study displays the observed effect sizes of the individual studies against the continuous variable sample size.

**Figure 11 ijerph-19-14280-f011:**
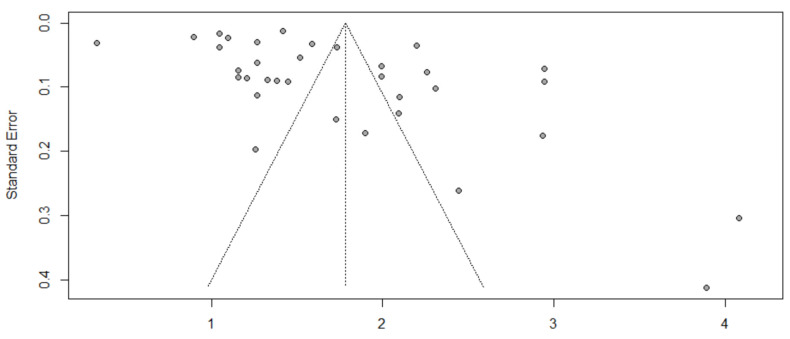
Funnel plot for the evaluation of potential publication bias. Each solid circle represents a study in the meta-analysis.

**Figure 12 ijerph-19-14280-f012:**
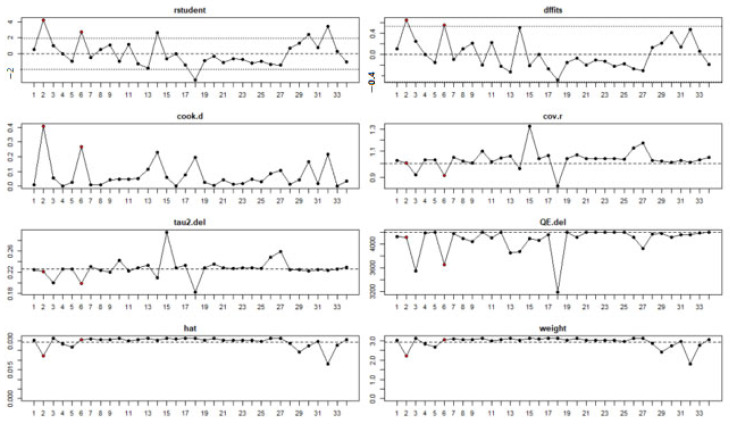
Baujat plot shows no single study that influenced the results.

**Table 1 ijerph-19-14280-t001:** Summary of the studies used in the systematic review and meta-analysis (n = 34).

Author	Reference	Year	Diabetes Prediction	Sample Size	Sensitivity (%)	Specificity (%)	Overall Classification Accuracy (%)	Classification Technique	Country First Author	Impact Factor
Rathmann et al.	[5]	2010	Prognostic	1353			88	LR	Germany	3.11
Upadhyaya et al.	[21]	2013	Diagnostic	663	97	99	98	NN	USA	5.45
Wang et al.	[6]	2013	Diagnostic	8640	87	79	90	ANN	China	3.24
Huang et al.	[7]	2015	Nephropathy	345	85	83	85	DT	China	3.24
Kuo et al.	[8]	2020	Diagnostic	149			78	DT	China	2.38
Pei et al.	[9]	2019	Prognostic	4205			95	DT	China	2.07
Casanova et al.	[10]	2016	Prognostic	2363			82	RF	USA	2.78
Rau et al.	[2]	2016	Risk factor analysis	2060	75	75	88	ANN	Taiwan	3.63
Ramezankhani et al.	[11]	2014	Prognostic	1995	31	98	91	DT	Iran	3.24
Ramezankhani et al.	[12]	2016	Prognostic	6647	70	79	78	DT	Iran	2.38
Ramezankhani et al.	[13]	2016	Prognostic	1164	22	99	91	DT	Iran	2.79
Dugee et al.	[14]	2015	Diagnostic	1018			76	LR	Mongolia & Finland	2.57
Esmaeily et al.	[15]	2018	Diagnostic	9528	71	70	71	RF	Iran	1.51
Heydari et al.	[22]	2016	Diagnostic	2536	98	67	95	DT	Iran	0.59
Nanri et al.	[23]	2015	Prognostic	37,416	84	80	80	LR	Japan	2.78
Cichosz et al.	[24]	2014	Diagnostic	5381			85	LR	Denmark	3.30
Olivera et al.	[25]	2017	Diagnostic	3709	66	69	74	ANN	Brazil	0.13
Upadhyaya et al.	[4]	2017	Prognostic	4208	99	99	58	Phenotyping	USA	0.00
Usharani & Shanthini	[26]	2021	Nephropathy	768			79	LR	India	4.59
Rodriguez-Romero et al.	[27]	2019	Nephropathy	6777			83	RF	USA	3.99
Parashar et al.	[28]	2014	Diagnostic	768			77	SVM	China	2.5
Farahmandian et al.	[29]	2015	Diagnostic	768			81	SVM	Iran	0.00
Khashei et al.	[30]	2012	Diagnostic	768			80	SVM	Iran	0.00
Bozkurt et al.	[31]	2014	Diagnostic	768	53	89	76	ANN	India	0.68
Kumari & Chitra	[32]	2013	Diagnostic	460			78	SVM	India	1.45
Anderson et al.	[33]	2016	Screening and diagnosis	9948	80	73	75	LR	USA	2.95
Alssema et al.	[34]	2011	Prognostic	18,301			74	LR	The Netherlands	7.11
Chen et al.	[35]	2015	Nephropathy	519			89	ANN	China	4.19
Marateb et al.	[36]	2014	Nephropathy	200	95	85	92	Hybrid model	Iran	3.43
Leung et al.	[37]	2013	Nephropathy	673			95	SVM	China	2.03
Chikh et al.	[38]	2012	Screening and diagnosis	768	85	92	89	CRISP	Algeria	3.06
Zheng et al.	[39]	2017	Screening and diagnosis	300			98	LR	China	3.03
Yu et al.	[40]	2016	Nephropathy	299	83	88	87	ANN	Taiwan	0.43
Meng et al.	[41]	2013	Risk factor analysis	1487	81	75	78	DT	China	1.74

**Table 2 ijerph-19-14280-t002:** Meta-analytic regression results (* implies significant value).

Model	β	SE	*p*-Values	Q_M_	df	*p*-Values
Publication year	−0.0359	0.0532	0.5001	0.4546	1	0.5001
Impact factor	0.1297	0.0809	0.1086	2.5747	1	0.1086
**Diabetes prediction**				2.2366	4	0.6923
Diagnostic	**Ref**					
Nephropathy	0.3391	0.3516	0.3348			
Prognostic	0.0219	0.3213	0.9456			
Risk factor analysis	−0.0210	0.5617	0.9701			
Screening and diagnosis	0.5811	0.4905	0.2361			
**Model types**				26.0392	8	0.0010
ANN	**Ref**					
CRISP method	0.3714	0.6090	0.5420			
Decision trees	0.2786	0.3035	0.3586			
Hybrid model	0.7166	0.6523	0.2720			
Linear regression	−0.1191	0.3047	0.6959			
Neural network	2.3564	0.6708	0.0004 *			
Phenotyping	−1.3977	0.5988	0.0196 *			
Random forest	−0.3935	0.3933	0.3171			
Support vector machine	−0.0946	0.3409	0.7813			

## Data Availability

Raw and processed data are available upon request to the corresponding author.

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
