# Peer review of "Accuracy of Machine Learning Classification Models for the Prediction of Type 2 Diabetes Mellitus: A Systematic Survey and Meta-Analysis Approach"

_ijerph, 2022, doi:10.3390/ijerph192114280_

Round 1

Reviewer 1 Report

This is an interesting and well-written paper. I have only some minor suggestions for improvement.

For discussion:

- The prevalence of Diabetes mellitus differs per race/ethnicity (see https://www.ncbi.nlm.nih.gov/pmc/articles/PMC5241767/). Is there a bias in your dataset (e.g. many white people), and how well would these results translate to other races/ethnicities? The same for gender.

- Line 166-168: "It was evident that publications related to applying ML techniques in diagnosing diabetes mellitus have increased significantly from 2013 to 2016. Based on the inclusion criteria, we noted a downward trend for publications in the past four years." What could be the reason for this? Is there limited trust in ML/AI algorithms? Or are the models more difficult to be improved?

Grammar/spelling/etc.:

- line 46: "DM" -> "Diabetes Mellitus"

- line 109: "impact factor of the articles" -> how were these impact factors calculated/obtained?

- line 146: explain CRISP method

- line 201: : "p-value < .0001)" -> remove the round bracket?

- line 266: "iImpact factor" -> "impact factor"

Reviewer 2 Report

This paper reviews recent soft-computing and statistical learning models in T2DM using a meta-analysis approach. They searched for papers using  soft-computing and statistical learning models focused on T2DM and published between 2010 to 2021 on three different search engines. The following issues should be addressed.

1. The authors can search for papers using  soft-computing and statistical learning models focused on T2DM and published between 2010 to 2022. This year also have some related work.

2.In table 1, we can see the indexes for different method are different, then the how can  they get the overall classification accuracy. They should give more detailed description.

3. The quality of Figure 1. Figure 5-8 should be high resolution. 

4.Compared to other models, the DT model performed the best. The authors should give the analysis for the reason.

Author Response

Please the attachment
